# Clinical Applications of the Molecular Landscape of Melanoma: Integration of Research into Diagnostic and Therapeutic Strategies

**DOI:** 10.3390/cancers17091422

**Published:** 2025-04-24

**Authors:** Imre Lőrinc Szabó, Gabriella Emri, Andrea Ladányi, József Tímár

**Affiliations:** 1Department of Dermatology, MTA Centre of Excellence, Faculty of Medicine, University of Debrecen, 4032 Debrecen, Hungary; szabo.imre.lorinc@med.unideb.hu (I.L.S.); gemri@med.unideb.hu (G.E.); 2HUN-REN-UD Allergology Research Group, University of Debrecen, 4032 Debrecen, Hungary; 3Department of Surgical and Molecular Pathology, National Institute of Oncology, 1122 Budapest, Hungary; ladanyi.andrea@oncol.hu; 4National Tumor Biology Laboratory, National Institute of Oncology, 1122 Budapest, Hungary; 5Department of Pathology, Forensic and Insurance Medicine, Semmelweis University, 1091 Budapest, Hungary

**Keywords:** cutaneous melanoma, melanoma genetics, immunotherapy, targeted therapy, adjuvant

## Abstract

Malignant melanoma was one of the first tumors to have its genetic background completed; however, few of the alterations became successful therapeutic targets. The knowledge on the immunobiology of melanoma led to the introduction of immune checkpoint inhibitors into clinical practice, but there is room for improvement concerning predictive markers of efficacy and resistance. In this review, we summarize the present status and outline key points for further development.

## 1. Introduction

According to the latest GLOBOCAN estimates, around 330,000 new melanoma patients were diagnosed in 2022, while the number of deaths from melanoma was approximately 58,000 [1]. This represents a major increase in incidence compared to the early 2000s (~130,000–160,000 new cases), with a smaller change in the annual mortality rate (approximately 40,000) [2,3]. These epidemiological data suggest that tumors are being detected at an early stage when surgical treatment is usually curative, and that advanced or metastatic tumors are being treated more effectively, thanks to the enormous progress made in this field over the last decade, one of the driving forces being molecular pathology.

## 2. Melanoma Genetics

The main cause of cutaneous melanoma is UV-B exposure (accounting for about 65%); increasing trends of recreational sun exposure such as sunbathing, combined with the trend toward reduced body surface coverage in modern beachwear, have contributed to a steady rise in melanoma incidence [4]. Pigmented nevi are very common but fortunately have a low tendency for malignant transformation. However, these benign lesions already carry the BRAF (v-Raf murine sarcoma viral oncogene homologue B1) exon 15/V600 mutation, which is also characteristic of melanoma, and a high number of nevi on the skin is associated with a higher risk of melanoma [5,6,7,8]. In superficially spreading melanomas, a characteristic anamnestic finding is a high incidence of nevi and intermittent UV exposure; also, the presence of the BRAF V600E mutation is common in this histological type [9]. However, melanoma can occur on non-UV exposed surfaces (acral regions, mucosal surfaces, or even in the eye): in these cases, the characteristic BRAF mutation is less frequent and the so-called UV-signature mutations are also not typical. The genetic consequence of UV exposition is that the genome-wide tumor mutational burden (TMB) in cutaneous melanomas is very high and approaches that of smoking-induced lung cancers [10] (Table 1).

Analysis of cutaneous melanoma data in The Cancer Genome Atlas has shown that the prevalence of BRAF mutations is ~45% (90% of which is V600E), followed by a prevalence of the NRAS oncogene exon 3/codon 61 of 15–20%, while other RAS mutations are less frequent (KRAS, HRAS) [7]. The mutation of the KIT (KIT-proto-oncogene receptor tyrosine kinase) gene, which is characteristic of mucosal melanomas, can be detected in melanomas of the skin at a much lower rate (~5%), involving more exons compared to GIST (gastrointestinal stromal tumors) [11]. The remaining melanomas can be described as triple-negative (triple-wild-type: BRAF, RAS, KIT).

The KIT/NRAS/BRAF signaling pathway is the dominant pathway in both melanocytes and melanomas, which is why mutations in these oncogenes occur with such frequency. However, not only oncogenic mutations occur in melanomas, but also those of suppressor genes, with a similar frequency (10–15%): TP53 (tumor protein p53), NF1 (neurofibromin 1), CDKN2A (cyclin-dependent kinase inhibitor 2 A), PTEN (phosphatase and tensin homolog) [5]. There is also a “rule of thumb” in melanoma, i.e., only one oncogene can be mutated in the same pathway, meaning that these three mutations (KIT/NRAS/BRAF) are mutually exclusive. Genome-wide variations may manifest as gene amplifications, affecting mainly oncogenes, but also genes of transcription factors, CCND1 (cyclin D1) and MITF (microphthalmia-associated transcription factor). On the other hand, copy number loss due to loss of heterozygosity is also common for suppressor genes. In addition to the dominant oncogenic and suppressor gene defects, other genetic abnormalities also occur in melanomas, making them quite heterogeneous. In BRAF-mutant melanomas, loss of the CDKN2A gene is typical, but amplification of MITF and CD274 (coding for PD-L1; programmed death ligand 1) is relatively common. Mutations in the Aurka inhibitor PPP6 C (protein phosphatase 6 catalytic subunit) gene are also typical in BRAF and NRAS mutant melanomas. IDH1 (isocitrate dehydrogenase 1) mutations are not frequent in melanomas (<10%) but may be of therapeutic relevance in later stages [5,12]. About half of the triple-wild-type melanomas have high TMB and are characterized by UV-induced base change (chronic UV exposure). These tumors are dominated by defects in oncosuppressor genes, with relatively high frequency of KRAS/NRAS amplification and mutations in the TERT (telomerase reverse transcriptase) promoter (Table 1) [13].

## 3. Hereditary Factors in Melanoma

As with all malignancies, melanoma has hereditary forms [14,15]. Familial melanoma with CDKN2A mutation (or loss of heterozygosity) has long been known. Interestingly, germline mutations can occur in pigmentation-related genes, such as the MITF transcription factor in melanocytes, MC1R (melanocortin receptor-1) regulated by MITF, the OCA2 gene (oculocutaneous albinism type-2), the SCL45A2 gene (solute carrier family 45 member 2) or even the tyrosinase or TYRP1 (tyrosinase-related protein-1) gene, which predispose to melanoma. On the other hand, germline mutations in CDK4 or in the DNA repair genes BAP1 (BRCA1 associated protein-1), APEX1 (apopurinic/apurinic-endodeoxyribonuclease 1) or even TERT-linked gene (TERT, POT1, ACD, TERF2IP) mutations may also predispose to melanoma. Interestingly, familial uveal melanoma is associated with germline mutations of other DNA repair genes, such as PALB2 (partner and localizer of BRCA2; involved in breast cancer), which leads to homologous recombination defects, or MLH1 (mutL homolog 1; part of Lynch syndrome), which leads to microsatellite instability (MSI) [16].

## 4. Predictive Genetic Markers

The molecular diagnosis of melanoma is based on the identification of BRAF-mutant tumors, but only BRAF V600E/K mutations are of therapeutic relevance. However, it does not matter whether the tumor is homozygous for this mutation (m/m) or heterozygous (m/wild). On the other hand, the frequency of the mutant allele may also be low due to clonal heterogeneity (m < 20%). However, it should be taken into account that significant clonal changes may occur during tumor progression and this cannot be predicted in advance; there may be a significant increase or decrease in the mutant clone, which may affect sensitivity to the targeted therapy, and therefore, mutation testing should be performed in metastatic tumors when possible [17]. As for KIT mutations, in contrast to GIST, exon 11/13 mutations are common in melanoma, potentially indicating therapeutic sensitivity, whereas exon 9 mutations are rare [11]. As for triple-wild melanoma, analysis with larger gene panels is worthwhile, as it has a higher probability to find targets for tumor-agnostic targeted therapies in this tumor type, such as RET, NTRK (neurotrophic tyrosine receptor kinase), ALK (anaplastic lymphoma kinase) or even IDH1 mutations [13].

We do not have routinely used predictive markers for immunotherapy, although many clinical trials have attempted to use PD-L1 TPS (tumor proportion score) for this purpose [18]. It should be noted that PD-L1 (CD274) amplified melanoma is certainly a tumor that may be hypersensitive to immunotherapy. The tumor-agnostic indication for immunotherapy is TMB-high tumors, which are very common in melanoma; even half of the so-called triple-wild-type melanomas belong to this category, in which case the use of immunotherapy is not questionable [13]. On the other hand, in melanoma, and this is true for all molecular subtypes, mismatch repair (MMR) deficiency and homologous repair deficiency (HRD) are very rare, if they occur at all; therefore, the indication of immunotherapy based on MSI is not possible, but the HRD variant of hereditary ocular melanoma is suitable for the use of poly (ADP-ribose) polymerase (PARP) inhibitors.

## 5. Up-to-Date Treatment of Cutaneous Melanoma

The treatment of cutaneous melanoma is based on prognostic classification, currently guided by the 8th edition of the American Joint Committee on Cancer (AJCC) TNM classification [15,19,20]. Accordingly, the first step in the case of clinical suspicion of melanoma is complete surgical excision and histological examination of the primary tumor. Prognostically, Breslow tumor thickness (to the nearest 0.1 mm) and the presence or absence of ulceration and microsatellites are the most important, but the histological description should include the histological type of melanoma, (lymphatic) vascular invasion, neurotropism/perineural invasion, tumor-infiltrating lymphocytes, regression, resection margins and mitotic rate, which will influence clinical decision-making [19].

In the case of ≥1 mm Breslow tumor thickness and clinically negative lymph node status, sentinel lymph node biopsy is performed. If possible, a wide excision of the primary tumor site is recommended in the same setting (1–2 cm margin of safety depending on tumor thickness) [20,21]. Sentinel node surgery for pT1 primary melanoma is not routinely indicated, but is indicated in the presence of pT1 b or other risk factors (e.g., high mitotic rate). For accurate staging, CT and/or PET/CT and cranial MR are recommended from stage IIB onwards, in addition to physical examination and lymph node ultrasound [15,21].

In high-risk primary melanoma (stage IIB/IIC), it is advisable to request molecular diagnostic testing, and in stage III/IV, it is essential [15,21]. The analysis should include at least testing for BRAF V600 mutations, and, if possible, NRAS, c-KIT and NTRK gene alterations [15,21]. Immunohistochemical analysis of PD-L1 expression is also recommended if there is a therapeutic implication, e.g., the indication for anti-PD-1 + anti-LAG-3 (nivolumab + relatlimab) combination in Europe is currently <1% PD-L1 expression on tumor cells. Additional molecular assays (e.g., TMB determination, gene expression pattern analysis, NGS, circulating tumor DNA detection), which are not currently part of clinical routine, will play an increasingly important role in therapeutic decision-making in the future once their predictive value in a therapeutic context is confirmed [22,23,24,25].

Recently, with the advent of biological therapies in adjuvant and more recently neoadjuvant indications, clinical practice in the management of melanoma is changing almost fundamentally. As an example, today, confirmation of sentinel lymph node metastasis is not routinely followed by radical lymph node block dissection or radiotherapy of the lymph node region, but by adjuvant systemic treatment for 1 year, starting within 12 weeks after surgery if indicated [20]. Furthermore, according to the latest ESMO recommendation, in the case of high-risk primary melanoma (pT3 b/pT4), omission of sentinel node surgery may be discussed with the patient if adjuvant treatment is available and can be administered [21].

Adjuvant radiotherapy is currently recommended for lentigo malignant melanoma after excision with an insufficiently wide safety zone, for desmoplastic or neurotropic melanoma if a pathological resection with ≥8 mm of intact margin cannot be ensured, or after surgical removal of extensive lymph node metastases [21].

### 5.1. Adjuvant Systemic Therapy

Patients with resected stage IIB/IIC melanoma may be given adjuvant anti-PD-1 (pembrolizumab or nivolumab) treatment after careful consideration of the benefit and risk [21]. In this group of patients, disease recurrence occurs within 3 years in at least one-third of cases, with distant metastasis in one-quarter of patients. Among patients receiving adjuvant immunotherapy, a reduction in the recurrence rate to one-quarter and in the rate of distant metastatic disease to one-sixth has been found (KEYNOTE-716) [26]. The impact on overall survival is not yet known.

Based on prognostic considerations, stage IIIA according to AJCC (8th edition) TNM classification and <1 mm metastatic size in the sentinel lymph node are not routine indications for adjuvant systemic treatment [27,28]. Apart from this, however, adjuvant drug therapy is recommended in resected stage III/IV melanoma, namely anti-PD-1 (pembrolizumab or nivolumab), or alternatively dabrafenib-trametinib in BRAF V600 mutant melanoma [20,29]. Adjuvant pembrolizumab treatment in stage III patients has an expected 7-year relapse-free survival of 50% (36% in the placebo group) and distant metastasis-free survival of 54% (42% in the placebo group) (EORTC1325/KEYNOTE-054) [30]. In stage IIIB/IIIC/IV (AJCC 7th edition) patients treated with adjuvant nivolumab after surgery, the expected 5-year relapse-free survival was 50%, distant metastasis-free survival 58% and overall survival 76% (CheckMate 238) [31].

In stage III BRAF V600 mutant melanoma, adjuvant dabrafenib-trametinib significantly improves relapse-free survival (10-year 48%, 32% in placebo group) and distant metastasis-free survival (10-year 63%, 48% in placebo group) (COMBI-AD) [32]. The clinical benefit is clear in the presence of BRAF V600E mutation, more uncertain in the presence of BRAF V600K mutation. Biomarker analysis suggested that patients with melanoma characterized by low TMB and high IFN-γ gene expression signature benefit most from adjuvant targeted therapy [32,33].

### 5.2. Neoadjuvant Therapy

Improved clinical outcomes can be achieved not only through new drugs with greater therapeutic efficacy or combinations of drugs that overcome resistance to therapy, but also by optimizing the timing of surgery and systemic treatment. Although only the results of clinical trials are known so far, neoadjuvant treatment as a therapeutic principle is expected to become part of daily practice in the near future [21,29,34]. One of the ways to achieve this is perioperative treatment. If the first three cycles of pembrolizumab treatment, planned for a 1-year duration, are given to a patient with stage IIIB-IV melanoma before metastasectomy and the remainder after, according to the results of a phase II trial, the expected event-free survival is significantly better than when the full treatment is given postoperatively (i.e., adjuvantly), but the effect on overall survival is questionable (SWOG S1801) [35]. The other concept is that a stage III patient receives two cycles of combined immunotherapy (nivolumab 3 mg/kg + ipilimumab 1 mg/kg) before lymph node surgery and adjuvant dabrafenib-trametinib (BRAF-mutant melanoma) or nivolumab after surgery, but only in the case of partial or absent pathological response. In the clinical trial, the odds of distant metastasis within 18 months were reduced from 38% to 14% (3% for complete/nearly complete pathological response) by neoadjuvant treatment compared with adjuvant nivolumab (NADINA). In addition, neoadjuvant nivolumab + ipilimumab treatment resulted in pathological complete or near-complete response in 59% of patients [36]. In a prospective study evaluating the clinical efficacy of perioperative nivolumab + relatlimab treatment in patients with resectable stage III–IV melanoma, two cycles of combination immunotherapy resulted in a similarly high (57%) pathological complete response rate [37]. However, appropriate patient selection is important, because delaying surgery may result in unresectability if there is no tumor response to systemic treatment [34]. For the same reason, combination immunotherapy as neoadjuvant treatment may be a better choice than monotherapy because of the higher chance of tumor response induction. Although BRAF + MEK inhibitor treatment also has a high probability of achieving complete pathological tumor regression, the expected relapse-free survival does not approach that achieved by immunotherapy [38].

### 5.3. Metastatic Disease

In unresectable metastatic melanoma, targeted (combined BRAF-MEK inhibitor) therapy and immune checkpoint blockade are the two most important types of treatment available (Table 2). These treatments differ substantially in terms of efficacy and side effects due to the different targets and mechanism. Furthermore, melanoma patients without BRAF mutations do not benefit from targeted therapy, whereas immunotherapy can be equally effective in all patients regardless of BRAF mutation status. In addition to these two basic considerations, relevant prognostic factors and limitations due to the side effect profile help to guide the choice of treatment in BRAF-mutant melanoma [20,21,29].

Initially, BRAF inhibitor monotherapy was available, but it was soon suggested that combining it with a MEK inhibitor could improve the side effect profile, efficacy and reduce resistance. Therefore, today, targeted therapy for BRAF mutations can only be given in combination, although if an adverse event is clearly associated with the MEK inhibitor (e.g., cardiomyopathy, ocular, pulmonary side effects), it is possible to continue BRAF inhibitor treatment as monotherapy. A higher proportion of patients with BRAF-mutant melanoma respond to combination BRAF-MEK inhibitor therapy than to immunotherapy (Table 2) and, more importantly, the positive effect is rapid. However, in terms of long-term survival, targeted therapy is far inferior to immunotherapy, because of the loss of efficacy due to the development of resistance that occurs sooner or later in the majority of patients. For this reason, international guidelines also recommend immunotherapy as a first-line treatment for BRAF-mutant melanoma [20,21,29]. However, BRAF-MEK inhibitor therapy should be used when a rapid effect is required and the slower-onset effect of immunotherapy may compromise short-term outcome [20,21].

The immunotherapeutic agents approved for the treatment of metastatic disease are pembrolizumab, nivolumab, nivolumab + ipilimumab, and nivolumab + relatlimab (Table 2). There is no routinely used predictive marker of efficacy, but there are clinical/pathological prognostic factors that influence treatment choice, as subgroup analyses of clinical trials suggest that immune checkpoint inhibitor combinations may in some cases provide a higher survival than monotherapy. For example, the combination of nivolumab + ipilimumab may be a better choice for asymptomatic brain metastases or acral melanoma in first-line settings, and also if the melanoma is BRAF mutant [20,28]. Clinical trial data suggest that when PD-L1 expression is <1% on tumor cells, the combination of nivolumab with relatlimab in first-line setting provides better progression-free survival compared to nivolumab monotherapy [43]. Anti-PD-1 monotherapy can provide survival equivalent to combined immune checkpoint inhibitor therapy in PD-L1 positive tumors and BRAF wild-type melanoma if the patient has no central nervous system metastases [28]. Immunotherapy can be effectively combined with radiotherapy for oligometastatic disease or oligoprogression, and stereotactic radiosurgery or fractionated stereotactic radiotherapy for melanoma brain metastases [21]. The risk of side effects should also be considered in the choice of therapy, as advanced age and comorbidities may make it more difficult to tolerate a therapy-associated side effect, the odds of which are particularly high with nivolumab + ipilimumab combination (55% grade 3–4) compared with nivolumab monotherapy (16.3% grade 3–4) [18]. The expected incidence of grade 3–4 adverse events with nivolumab + relatlimab is 21.1% [43].

Data on the optimal treatment strategy for unresectable metastatic melanoma diagnosed after adjuvant treatment are currently scarce [21,28]. In metastatic melanoma, after progression on (partially adjuvant) anti-PD-1 monotherapy, objective response rates of 29% with pembrolizumab + ipilimumab combination, and 28% with nivolumab + ipilimumab combination have been reported [49,50]. In a phase I clinical trial with cemiplimab (anti-PD-1) + fianlimab (anti-LAG-3), the objective response rate in metastatic melanoma was 63% in patients who had not received prior anti-PD-1 treatment, 61.5% in patients who had received prior adjuvant anti-PD-1 treatment and 13.3% in patients who had received prior anti-PD-1 treatment for advanced melanoma [51]. These data point towards immune checkpoint inhibitor combinations for progression following adjuvant anti-PD-1 treatment. In BRAF-mutant melanoma, progression during or within 6 months of adjuvant anti-PD-1 therapy is associated with a higher chance of immunotherapy resistance, and therefore the initiation of a BRAF + MEK inhibitor combination should be considered [28].

In BRAF-mutant metastatic melanoma, treatment with nivolumab + ipilimumab appears to be more favorable for survival than first-line BRAF + MEK inhibitor [21,28]. However, in the presence of very high LDH levels, poor general health, liver metastases or symptomatic brain metastases (requiring >4 mg dexamethasone), first-line BRAF + MEK inhibitor treatment may be justified, given the rapid tumor regression expected from targeted therapy in responding patients [28]. It has been suggested that the chance of progression-free survival in the presence of elevated LDH and liver metastases may be increased by the administration of targeted therapy for 3 months prior to immunotherapy, but clinical trial data are still scarce [21,28]. If progression occurs after first-line immune checkpoint inhibitor therapy in BRAF-mutant unresectable melanoma, a second-line BRAF + MEK inhibitor combination may be recommended [21,29]. If progression occurs on first-line targeted therapy, second-line immune checkpoint inhibitor administration of anti-PD-1 ± anti-LAG-3 is an option [29]. In the case of repeated progression, however, a “rechallenge”, i.e., restarting the BRAF + MEK inhibitor combination, may be attempted [52].

Besides the above-described therapeutic modalities, a new option was recently included in the treatment armamentarium of metastatic melanoma: lifileucel, a centrally manufactured, autologous tumor-infiltrating lymphocyte (TIL) product, was approved by the FDA in 2024 for patients who progressed after immune checkpoint inhibitor or targeted therapy, making it the first cellular therapy approved for solid tumor. The approval was based on the phase II C-144–01 trial, showing 31.4% objective response rate in a cohort of heavily pretreated patients [53]. The results are encouraging; however, similarly to other personalized cell-based modalities, TIL therapy is labor-intensive and logistically more challenging than off-the-shelf drugs [54].

A further therapeutic option for unresectable metastatic melanoma is the intralesional treatment with talimogene laherparepvec (T-VEC), a genetically modified oncolytic Herpes simplex virus, which can be applied in case of injectable (cutaneous/subcutaneous or nodal) metastases [55]; it is suggested when other therapy regimens are contraindicated [29].

## 6. Molecular Basis of Therapy Resistance

### 6.1. Targeted Therapy

Arguments against BRAF +/− MEK inhibitor therapies were that the duration and depth of response is generally much shorter than in the case of immunotherapies and progression develops rapidly. Genetic studies have shown that in such tumors, similar to other targeted therapies, new so-called resistance mutations develop in molecular target genes (BRAF, MEK1/2). One reason for this may be that the proportion of tumor cells carrying the gene defect in the tumor to be treated is not taken into account; therefore, the majority of tumors are usually not dominated by cells carrying the gene defect, so there is a high chance of the growth of alternative clones which are resistant to treatment. It has been observed that mutations in certain elements of the AKT signaling pathway (PIK3CA, PIK3R1/2, AKT1) are frequently expressed in resistant tumors, and that PTEN mutant tumors have primary resistance to BRAF inhibitor therapies or it may develop in response to treatment [56,57,58]. PTEN, in turn, is an important inhibitor of PIK3CA, so loss of its function may provide unrestricted activity of this signaling pathway. Experimental data suggest that in BRAF/PI3K/PTEN mutant tumors, there is potential for a successful combination of a BRAF inhibitor and a PI3K inhibitor [59]. In breast cancer, such inhibitors have been successfully used in case of hormone resistance development [60]. The main genetic alterations contributing to targeted therapy resistance in melanoma are listed in Table 3.

### 6.2. Immunotherapy

As discussed above, immune checkpoint inhibitor therapies have no routinely used, validated predictive markers in melanoma. In comparison, these therapies are effective in about half of melanoma patients in the short term and are useful in about one-third of patients in the long term. One of the molecular alterations found to be associated with primary and acquired resistance to immune checkpoint blockade is loss of the tumor suppressor PTEN [5,61,62]. Another genetic feature leading to immunotherapy resistance in melanoma is the loss of HLA-I, which may be caused by the loss of the corresponding region of chromosome 6 [63], or the mutation or loss of heterozygosity (LOH) of the β2-microglobulin (B2M) gene [64,65,66,67] (Table 3). The reason is that recognition and destruction of tumor cells by specific cytotoxic T lymphocytes require not only the tumor antigen but also the HLA molecule that presents it. However, such mutations are relatively rare; HLA class I expression loss is more frequently caused by epigenetic mechanisms [68] and determining protein expression may be more useful from functional aspects. HLA-I protein expression was found to be associated with the efficacy of the anti-CTLA-4 antibody ipilimumab in metastatic melanoma patients [69,70]. In a small cohort of ipilimumab-treated melanoma patients, a decrease in HLA-I expression was demonstrated in progressing metastases of nonresponding patients [71].

Another frequently described mode of resistance to immune checkpoint inhibitors is damage to the IFN-γ signaling pathway, which may be caused by mutation of the JAK1/2 (Janus kinase 1/2) gene [64,65] (Table 3). Based on this, an IFN-γ-related mRNA expression gene panel has been developed and has been shown to be a predictive marker for immunotherapy [72]. IFN-γ plays a pivotal role in antitumor immune response, partly through increasing the expression of MHC molecules and thus antigen presentation. However, besides IFN-γ (class II IFN), class I interferons (IFN-α and -β) also have immunological effects and are implicated in the efficacy of immunotherapy [73]. In the past, type I IFN treatment was an important tool for adjuvant treatment of high-risk melanoma, but it has proved to have modest efficacy [74]. However, this fact may be of importance in the late progression of melanomas in these patients. According to a genomic copy number variation study, patients with progressive tumors previously treated with type I IFN have a characteristic genetic fingerprint, which may be important for the efficacy of subsequent immunotherapy [75]. Genomic studies have also revealed that during metastatic progression of melanoma, metastases display genetic features that are characteristic of immune cells, not only PD-L1 (CD274) but also CTLA4 and several other genes that may be important for immune resistance, constituting a newly identified form of phenotypic plasticity, the so-called immunogenic mimicry [76].

## 7. New Therapies on the Horizon

### 7.1. The Problem of Triple-Wild (BRAF/NRAS/KIT) Melanomas

In triple-wild-type melanomas, great attention should be paid to the so-called rare gene defects for which there may be a tumor-agnostic drug indication. These gene defects may co-occur with those of melanoma driver genes, but in that case, their biological and therapeutic significance is small because the driver genes suppress the activity of these mutations. In other words, in triple-wild melanoma, it is necessary to investigate the otherwise very rare mutations/fusions, for which inhibitors (approved in other tumors) are available, such as those affecting NTRK1/3 genes [77] or RET [78]. In a Hungarian triple-wild-type melanoma cohort, such mutations were detected in half of the cases [13].

On the other hand, these tumors would be ideal immunotherapy targets. However, it should be noted that only half of these tumors have a high TMB. Our analysis also showed that these TMB-high tumors also harbored immunosensitizing gene defects (CSMD1, TTN, MUC16) [79,80,81], further supporting the use of immune checkpoint inhibitors [13]. At the same time, an interesting finding was that in the low TMB melanomas, a gene defect associated with immunotherapy sensitivity was also present: AHNAK2 [82]. Based on these findings, it is suggested that in triple-wild-type melanomas, it is advisable to perform a large gene panel NGS study, including possible fusion genes but also determining TMB, as this will increase the likelihood of detecting targetable mutations.

### 7.2. gp100 Immunotherapy

For decades, the pathological diagnosis of melanoma has been using glycoprotein 100 (gp100), a product of the PMEL gene and a protein of the melanosome membrane, as a marker, and the HMB45 antibody is used to detect it. In the early 2000s, gp100 was already emerging as a therapeutic target, using gp100 peptides for vaccination, but these have not been effective in clinical trials [83]. More recently, tebentafusp, belonging to the new class of immunotherapy agents named T cell engagers or ImmTACs (immune-mobilizing monoclonal T cell receptors against cancer), was developed [84]. Tebentafusp is a bispecific antibody composed of a single-chain antibody fragment recognizing the T cell antigen CD3, and a T cell receptor that recognizes the gp100 peptide (YLEPGPVTA) bound to HLA-A*02:01. This construction enables redirecting the T cells to target cells carrying the antigen peptide in presented form, resulting in polyclonal activation of T cells irrespective of their original specificity, cytokine release and melanoma cell death [85]. Clinical trials have been conducted in patients with metastatic uveal melanoma [86,87,88,89]. The proven survival benefit of tebentafusp treatment has led to the approval of the drug for this indication. A prerequisite for its use is the HLA-A*02:01 haplotype, which must be determined from peripheral blood. To date, the prevalence of this haplotype is known to be approximately 50% in Caucasian patients. Unusually, in cancer therapy, the assessment of tumor response according to Response Evaluation Criteria in Solid Tumors (RECIST), version 1.1, does not necessarily provide a good estimate of the overall survival benefit of tebentafusp treatment [86,87,88,89]. The most common adverse effects of treatment are cytokine-mediated ones, such as cytokine release syndrome, and also skin-related ones, such as rash or pruritus [86,87,88,89]. A characteristic side effect is vitiligo and the disappearance of nevi, as the antigen is also expressed on melanocytes. The drug has shown promising activity also in cutaneous melanoma refractory to standard immunotherapy, based on phase I/II clinical trials applying it as monotherapy or in combination with immune checkpoint inhibitors [85,90,91]. This suggests that this novel immunotherapy may also be effective in cutaneous (or mucosal) melanomas.

### 7.3. mRNA and Neoantigen Vaccines

COVID-19 vaccination has demonstrated that mRNA-based vaccines have clinical relevance. Nevertheless, successful vaccination against tumors is much more challenging. Over the decades, anti-melanoma vaccines have been developed and tested on the basis of many different concepts. One approach of applying mRNA in vaccination has been the transfection of autologous monocyte-derived dendritic cells with mRNA of melanoma antigens, and delivering them back to patients with advanced melanoma in the form of cell therapy. One of the first such vaccination procedures used gp100 and tyrosinase as tumor antigens, resulting in the induction of antitumor immunological responses, but clinical responses were rare, even in case of combination with platinum-based chemotherapy [92,93]. In another clinical trial, dendritic cells were loaded with four melanoma antigens and three immune cell-stimulating antigens and combined with ipilimumab: the overall response rate was 38% [94]. These approaches were promising, but like all cell therapies, they were very expensive and difficult to implement.

In the case of direct injection of mRNA, various approaches were used to circumvent the problem of the inherent instability of mRNAs [95]. In an early phase I/II trial, patients with metastatic melanoma were treated intradermally with vaccines containing six melanoma antigens (MelanA, tyrosinase, gp100, MAGEA1/3, survivin), applying RNA protection with the cationic peptide protamin [96]. More recent studies use mRNA-loaded nanoparticles, similarly to the COVID-19 mRNA vaccines [95,97]. The most successful new approach was pre-sequencing melanomas and selecting up to 34 genes encoding predicted neoantigens, then applying it as a personalized vaccine in combination with pembrolizumab to treat stage IIIB-IV melanoma patients adjuvantly (phase IIb study, KEYNOTE-942) [98]. Vaccination increased the 18-month RFS to 79% compared to 62% in the pembrolizumab arm, leading to initiation of phase III testing of the vaccine. Personalized neoantigen vaccines are also applied using alternative delivery methods (e.g., peptides or viral vectors) in early phase trials [99,100].

### 7.4. Inhibition of Mutant NRAS

The second most frequently mutated oncogene in melanoma is NRAS, but unlike mutant KRAS, the gene defect affects the Q61 codon of exon 3 and selective inhibitors have not yet been developed. NRAS mutant melanomas are also characterized by the activation of several signaling pathways (PI3K/AKT, RAL-GEF and cell cycle regulators) in addition to the MAPK pathway, in contrast to BRAF mutants. This may explain why the MEK inhibitor binimetinib has not been shown to be effective in NRAS mutant melanomas when used as monotherapy [101]. It has also been observed that this ineffectiveness occurs because of a specific feedback loop that involves BRAF/CRAF in the growth factor receptor pathway. MEK inhibitors should not be combined with AKT inhibitors because they are ineffective in NRAS mutant melanoma. At the same time, binimetinib has been shown to be effective when combined with a CDK4 inhibitor (ribociclib), especially when CDK4 or cyclin D1 mutations were present in the tumor (overall response rate 32%) [102]. It is important to note that the CDK4 inhibitor was administered at a lower dose than is usual in breast cancer. The combination of the MEK inhibitor with a pan-RAF inhibitor seems more rational. In a phase I trial in pretreated NRAS mutant melanoma patients, the pan-RAF inhibitor naporafenib was combined with trametinib and an objective response rate of 46% was achieved [103]. These results are encouraging, but a real breakthrough can only be expected from new NRAS inhibitors, which are being developed with great effort.

### 7.5. New Combinatorial Approaches Involving ICIs and Other Agents

Various combinations of immune checkpoint inhibitors and agents with different mechanisms of action have been explored both in preclinical investigations and in clinical trials. For example, promising results were obtained using PD-1/PD-L1 inhibitors in combination with blocking other inhibitory immune checkpoints such as TIM-3 or TIGIT, agonists of costimulatory receptors as OX40 (CD134) and CD137, adoptive cell therapies, oncolytic viruses, or the above-mentioned mRNA vaccines [98,104]. Another recent approach involves targeting the androgen receptor, proved to contribute to melanoma aggressiveness and immune escape [105,106], in combination with PD-1 blocking [107].

## 8. Conclusions

Malignant melanoma of the skin is a typical example showing that our knowledge on genetics (BRAF is a driver oncogene) and biology (it is an immunosensitive cancer) was sufficient to fundamentally change clinical practice. However, our recent knowledge on the genetics of cutaneous melanoma is far more complex and identified actionable gene defects such as NRAS, KIT or IDH1 mutations and NRAS mutant melanoma are now in clinical trials. Furthermore, the genomics of target therapy resistance are also known but have not been used for novel innovative therapies such as those targeting PIK3CA or AKT1 mutations. On the other hand, predictive markers of immunotherapies have been identified and validated, such as IFN-signature, but not used in everyday clinical practice yet. On the other hand, resistance markers are well known, such as HLA-I and/or B2M loss, but again, have not been validated for efficacy. In the meantime, novel types of immunotherapies are emerging, targeting gp100 melanoma antigen, or several melanoma antigens by mRNA vaccines. Last but not least, personalized genomic knowledge of the individual melanoma obtained from NGS (next generation sequencing) can now be exploited to design personalized mRNA vaccines. Accordingly, these new directions of clinical developments may change the therapeutic landscape of cutaneous melanoma in the near future.

## Figures and Tables

**Table 1 cancers-17-01422-t001:** Clinically relevant gene defects of melanoma.

	**Sporadic**		**Inherited**	
	Prognostic	Predictive	Prognostic	Predictive
**Cutaneous** **melanoma**			CDKN2/p16m MC1Rm OCA2m TYRm TYRP1m TERTm BAP1m CDK4m APEX1m	
Target therapy		BRAFm NRASm KITm IDH1m		
Immunotherapy		PDL1amp TMB-high		
	**Sporadic**		**Inherited**	
	Prognostic	Predictive	Prognostic	Predictive
Tumor-agnostic indications		NTRKf RETf ROS1f ALKf		
**Mucosal** **melanoma**			?	?
Target therapy		KITm BRAFm		
Immunotherapy		TMB-high		
**Ocular** **melanoma**	GNAQm GNA1m		MLH1m PALB2m	
Target therapy		BRAFm		HRD PALB2m
Immunotherapy		TMB-high		MLH1m MSI TMB-high

Abbreviations: amp: amplification, f: fusion, HRD: homologous recombination deficiency, m: mutation, MSI: microsatellite instability, TMB: tumor mutation burden.

**Table 2 cancers-17-01422-t002:** Treatment of cutaneous melanoma—systemic treatment options of metastatic or unresectable disease.

	Dosing ^#^	Response Rate	Overall Survival
Immune checkpoint inhibitors			
Pembrolizumab (anti-PD-1) [39]	200 mg i.v. every 3 weeks or 400 mg i.v. every 6 weeks	45.8%	10 year: 34%
Nivolumab (anti-PD-1) [40]	240 mg i.v. every 2 weeks or 480 mg i.v. every 4 weeks	42%	10 year: 37%
Nivolumab + ipilimumab * (anti-PD-1 + anti-CTLA-4) [40]	1 mg/kg + 3 mg/kg i.v. every 3 weeks *	50%	10 year: 43%
Nivolumab + ipilimumab * (anti-PD-1 + anti-CTLA-4) [41,42]	3 mg/kg + 1 mg/kg i.v. every 3 weeks *	45.6%	3 year: 59%
Nivolumab + relatlimab (anti-PD-1 + anti-LAG-3) [43]	480 mg + 160 mg i.v. every 4 weeks	43.1%	3 year: 55.8%
Combined BRAF + MEK inhibitor target therapy (in case of BRAF V600E/K mutation)			
Dabrafenib + trametinib [44]	2 × 150 mg + 1 × 2 mg daily p.o.	68%	5 year: 34%
Vemurafenib + cobimetinib ^#^ [45]	2 × 960 mg + 1 × 60 mg daily p.o. **	70%	5 year: 31%
Encorafenib + binimetinib [46]	1 × 450 mg + 2 × 45 mg daily p.o.	64.1%	7 year: 27.4%
cKIT inhibitor target therapy (in case of cKIT mutation)			
Imatinib [47]	1 × 400 mg daily p.o.	21.8%	2 year: 29.5%
MEK1/2 inhibitor target therapy (in case of NRAS mutation)			
Binimetinib [48]	2 × 45 mg daily p.o.	15%	-

^#^ according to the approved dosage, * 4 cycles, followed with nivolumab maintenance according to the monotherapy above, ** cobimetinib in 3-week cycles, followed with 1-week pause. Abbreviations: p.o.: per os, i.v.: intravenously.

**Table 3 cancers-17-01422-t003:** Genetic basis of therapy resistance of cutaneous melanoma.

Therapy	Most Common Genetic Alterations
BRAF (+MEK) inhibitors	PTENm BRAFrm MEK1/2rm PIK3CAm PIK3R1/2m AKT1m
Immunotherapy (anti-PD-1/anti-PD-L1/anti-CTLA-4)	PTENm HLA-I loss B2M loss B2Mm JAK1/2m

m: mutation, rm: secondary resistance mutation.

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
