# Peer review of "Clinical Applications of the Molecular Landscape of Melanoma: Integration of Research into Diagnostic and Therapeutic Strategies"

_cancers, 2025, doi:10.3390/cancers17091422_

Round 1

Reviewer 1 Report

Comments and Suggestions for Authors

This review article focuses on the clinical applications of the molecular landscape of melanoma. The authors provide a comprehensive summary of the current knowledge regarding this topic, including the genetic aspects of melanoma, up-to-date treatment strategies for cutaneous melanoma, the molecular basis of therapy resistance, and emerging new therapies. Overall, the paper is well-written and highly informative, without any significant concerns. I believe it will be of great interest to the journal's readers. One minor concern is that the use of too many abbreviations detracts from readability. I suggest omitting abbreviations that appear only once in the text.

Author Response

This review article focuses on the clinical applications of the molecular landscape of melanoma. The authors provide a comprehensive summary of the current knowledge regarding this topic, including the genetic aspects of melanoma, up-to-date treatment strategies for cutaneous melanoma, the molecular basis of therapy resistance, and emerging new therapies. Overall, the paper is well-written and highly informative, without any significant concerns. I believe it will be of great interest to the journal's readers. One minor concern is that the use of too many abbreviations detracts from readability. I suggest omitting abbreviations that appear only once in the text.

Response

We are very pleased that you have a positive opinion on our review paper.  We have corrected the Abbreviation section (and the text) according to the suggestion.

Reviewer 2 Report

Comments and Suggestions for Authors

Dear authors, I suggest minor revisions.

Tabel 1 could be graphical improved? Or it could be enriched of details?

I suggest also to expand conclusion chapter.

I add more specific comments on the review on cutaneous melanoma I reviwed:
• As authors write in abstract chapter , this review aims to bridge the perspectives of researchers studying the molecular background of melanoma and clinicians treating patients with melanoma by providing an overview of the genetic alterations relevant to the development and progression of melanoma, the latest therapeutic recommendations and treatment gaps. Their goal could be the improve of understanding and management of cutaneous melanoma.
• I consider a very good and updated review but as review it not add something in particular to the field. 
• Authors consider very well that researchers worked very hardly on immunotherapy of melanoma in the last 20 years but not all data were used in onology treatments,  so they want to explain the perspective on melanoma treatements for the future.
• In the conclusion they consider new biomarkers the future of cancer therapy, and it is conceptually correct, but in their manuscript they don't have a detailed chapter about this topic. So I suggested in my response to authors to add details in conclusion chapter.
• References are appropriated, updated and well representative of international landscape.
• I suggested to improve Table 1. I suggested to improve for graphics and details. 

Thank you

Author Response

Dear authors, I suggest minor revisions.

Tabel 1 could be graphical improved? Or it could be enriched of details?

I suggest also to expand conclusion chapter.

I add more specific comments on the review on cutaneous melanoma I reviwed:
• As authors write in abstract chapter , this review aims to bridge the perspectives of researchers studying the molecular background of melanoma and clinicians treating patients with melanoma by providing an overview of the genetic alterations relevant to the development and progression of melanoma, the latest therapeutic recommendations and treatment gaps. Their goal could be the improve of understanding and management of cutaneous melanoma.
• I consider a very good and updated review but as review it not add something in particular to the field. 
• Authors consider very well that researchers worked very hardly on immunotherapy of melanoma in the last 20 years but not all data were used in onology treatments,  so they want to explain the perspective on melanoma treatements for the future.
• In the conclusion they consider new biomarkers the future of cancer therapy, and it is conceptually correct, but in their manuscript they don't have a detailed chapter about this topic. So I suggested in my response to authors to add details in conclusion chapter.
• References are appropriated, updated and well representative of international landscape.
• I suggested to improve Table 1. I suggested to improve for graphics and details. 

Responses

Thank you for the constructive suggestion to improve our review. 

As suggested we have re-shaped  Table1. and made it more reasonable.

As it was suggested we have expanded the Conclusion chapter and included comments on biomarkers for target- and immunotherapies.

We have also compiled a new Table 3. on biomarkers of resistance since that issue was described in Chapter 6. Molecular basis of therapy resistance.

Reviewer 3 Report

Comments and Suggestions for Authors

The Review by Dr Imre LÅ‘rinc Szabó et al. describes a general overview of the current state of research on cutaneous melanoma, particularly focusing on genetic alterations, immunotherapy, and treatment advancements. The topic is intriguing, however I have different Major concerns:

  • Summary: The authors should consider adding a sentence about the specific gaps or challenges in current melanoma treatments (e.g., resistance to therapy, side effects, or patients who do not respond).
  • The Abstract is well-structured, but some sentences are a bit dense and could be simplified for better readability.
  • The Simple Summary and Abstract sections introduce important topics, but both could benefit from more concrete examples or details. This will strengthen the article's appeal to a broader audience, including clinicians and researchers, by providing clarity on what specific genetic alterations are involved, which therapeutic strategies are being explored, and where the gaps in knowledge or treatment remain.
  • The review is interesting, but in general, it doesn't add much new information. The authors could consider adding a paragraph on new combinatorial approaches that involve immune checkpoint inhibitors (ICIs) and drugs targeting other intracellular markers. In this context, the role of steroid hormone receptors and generally considered hormone-insensitive tumours is increasingly emerging. Therefore, consider combinatorial approaches targeting the androgen receptors (by genetic or pharmacological approaches) and ICIs: https://www.nature.com/articles/s41419-025-07350-4 (AR knockdown improves the effects of anti-PD1, anti-PDL1, anti-CTLA4 drugs in preclinical models); doi: 10.1038/s41467-023-42239-w  after all, also BRAF/MEK inhibitors works well if combined with AR Antagonists; doi: 10.1084/jem.20201137; https://doi.org/10.1158/1078-0432.CCR-22-2812

Comments on the Quality of English Language

The English language should be improved

Author Response

The Review by Dr Imre LÅ‘rinc Szabó et al. describes a general overview of the current state of research on cutaneous melanoma, particularly focusing on genetic alterations, immunotherapy, and treatment advancements. The topic is intriguing, however I have different Major concerns:

  • Summary: The authors should consider adding a sentence about the specific gaps or challenges in current melanoma treatments (e.g., resistance to therapy, side effects, or patients who do not respond).
  • The Abstract is well-structured, but some sentences are a bit dense and could be simplified for better readability.
  • The Simple Summary and Abstract sections introduce important topics, but both could benefit from more concrete examples or details. This will strengthen the article's appeal to a broader audience, including clinicians and researchers, by providing clarity on what specific genetic alterations are involved, which therapeutic strategies are being explored, and where the gaps in knowledge or treatment remain.
  • The review is interesting, but in general, it doesn't add much new information. The authors could consider adding a paragraph on new combinatorial approaches that involve immune checkpoint inhibitors (ICIs) and drugs targeting other intracellular markers. In this context, the role of steroid hormone receptors and generally considered hormone-insensitive tumours is increasingly emerging. Therefore, consider combinatorial approaches targeting the androgen receptors (by genetic or pharmacological approaches) and ICIs: https://www.nature.com/articles/s41419-025-07350-4 (AR knockdown improves the effects of anti-PD1, anti-PDL1, anti-CTLA4 drugs in preclinical models); doi: 10.1038/s41467-023-42239-w  after all, also BRAF/MEK inhibitors works well if combined with AR Antagonists; doi: 10.1084/jem.20201137; https://doi.org/10.1158/1078-0432.CCR-22-2812

Responses

The Review by Dr Imre LÅ‘rinc Szabó et al. describes a general overview of the current state of research on cutaneous melanoma, particularly focusing on genetic alterations, immunotherapy, and treatment advancements. The topic is intriguing, however I have different Major concerns:

  • Summary: The authors should consider adding a sentence about the specific gaps or challenges in current melanoma treatments (e.g., resistance to therapy, side effects, or patients who do not respond).
  • The Abstract is well-structured, but some sentences are a bit dense and could be simplified for better readability.
  • The Simple Summary and Abstract sections introduce important topics, but both could benefit from more concrete examples or details. This will strengthen the article's appeal to a broader audience, including clinicians and researchers, by providing clarity on what specific genetic alterations are involved, which therapeutic strategies are being explored, and where the gaps in knowledge or treatment remain.
  • The review is interesting, but in general, it doesn't add much new information. The authors could consider adding a paragraph on new combinatorial approaches that involve immune checkpoint inhibitors (ICIs) and drugs targeting other intracellular markers. In this context, the role of steroid hormone receptors and generally considered hormone-insensitive tumours is increasingly emerging. Therefore, consider combinatorial approaches targeting the androgen receptors (by genetic or pharmacological approaches) and ICIs: https://www.nature.com/articles/s41419-025-07350-4 (AR knockdown improves the effects of anti-PD1, anti-PDL1, anti-CTLA4 drugs in preclinical models); doi: 10.1038/s41467-023-42239-w  after all, also BRAF/MEK inhibitors works well if combined with AR Antagonists; doi: 10.1084/jem.20201137; https://doi.org/10.1158/1078-0432.CCR-22-2812

Round 2

Reviewer 3 Report

Comments and Suggestions for Authors

It Is suitable for publication